# Glucocorticoids Equally Stimulate Epithelial Na^+^ Transport in Male and Female Fetal Alveolar Cells

**DOI:** 10.3390/ijms21010057

**Published:** 2019-12-20

**Authors:** Mandy Laube, Diana Riedel, Benjamin Ackermann, Melanie Haase, Ulrich H. Thome

**Affiliations:** 1Department of Pediatrics, Division of Neonatology, University of Leipzig, 04103 Leipzig, Germany; 2Division of Neonatology, University Hospital for Children and Adolescents, Women’s and Children’s Hospital, 04103 Leipzig, Germany

**Keywords:** lung, glucocorticoids, sex differences, ENaC, preterm infants

## Abstract

Preterm infants frequently suffer from respiratory distress syndrome (RDS), possibly due to lower expression of epithelial Na^+^ channels (ENaC). RDS incidence is sex-specific, affecting males almost twice as often. Despite the use of antenatal glucocorticoids (GCs), the sex difference persists. It is still controversial whether both sexes benefit equally from GCs. We previously showed that Na^+^ transport is higher in female compared with male fetal distal lung epithelial (FDLE) cells. Since GCs increase Na^+^ transport, we hypothesized that their stimulating effect might be sex-specific. We analyzed FDLE cells with Ussing chambers and RT-qPCR in the presence or absence of fetal serum. In serum-free medium, GCs increased the ENaC activity and mRNA expression, independent of sex. In contrast, GCs did not increase the Na^+^ transport in serum-supplemented media and abolished the otherwise observed sex difference. Inhibition of the GC receptor in the presence of serum did not equalize Na^+^ transport between male and female cells. The GC-induced surfactant protein mRNA expression was concentration and sex-specific. In conclusion, female and male FDLE cells exhibit no sex difference in response to GCs with regard to Na^+^ transport, and GR activity does not contribute to the higher Na^+^ transport in females.

## 1. Introduction

Pulmonary adaptation to air breathing is an indispensable step to extra-uterine survival. During fetal lung development, the lung is filled with fluid that has to be removed prior to birth to enable postnatal lung function. Alveolar fluid clearance (AFC), is driven by epithelial Na^+^ transport accomplished by epithelial Na^+^ channels (ENaC) in the apical membrane compartment and the Na,K-ATPase in the basolateral membrane compartment of alveolar epithelial cells. Apical Na^+^ entry followed by basolateral Na^+^ extrusion creates an osmotic driving force causing fluid absorption from the airspaces into the interstitium and the bloodstream.

Respiratory distress is a frequent complication of premature birth due to structural and functional lung immaturity. In addition to surfactant deficiency, premature infants exhibit a decreased AFC [1], possibly due to lower expression of Na^+^ channels [2]. Therefore, insufficient AFC contributes to the development of the respiratory distress syndrome (RDS) [2]. Noteworthy, the incidence of RDS shows a sex ratio of 1:1.7 [3,4,5], with males developing RDS almost twice as often compared with female infants of the same gestational age. Consequently, male mortality exceeds female mortality during the first week of life [6,7]. Although this “male disadvantage” has been known for more than 40 years, male sex remains an independent risk factor for RDS development [3,4]. Our group recently demonstrated sex-specific differences in alveolar epithelial Na^+^ transport [8]. Male sex was associated with decreased Na^+^ transport and lower *ENaC* and *Na,K-ATPase* expression in fetal distal lung epithelial (FDLE) cells [8]. ENaC consists of three homologous subunits, α-, β- and γ-ENaC [9], and the Na,K-ATPase is composed of α_1_- and β_1_-subunits in alveolar type II (ATII) cells [10], which all exhibited lower mRNA expression in males. This was accompanied by higher lung fluid levels in male fetuses compared with females, suggesting lower or delayed onset AFC in males. Furthermore, estradiol and progesterone increased Na^+^ transport and Na^+^ channel expression especially in females, while estrogen receptor inhibition decreased Na^+^ channel expression and eliminated the sex difference [11]. Testosterone lacked any effect on Na^+^ transport in male and female FDLE cells, and flutamide, an androgen receptor antagonist, did not affect the gender imbalance, suggesting that Na^+^ transport is largely unaffected by androgens [11].

Antenatal steroid treatment reduces the incidence and severity of RDS in preterm infants. It is routinely administered in women at risk of preterm birth to improve perinatal survival and lung function [12]. Antenatal glucocorticoids (GCs) accelerate late-gestation lung maturation in low doses by enhancing surfactant synthesis, increasing the volume density of ATII cells and upregulating AFC [13,14,15]. Mice lacking intracellular glucocorticoid receptors (GR) died of respiratory failure shortly after birth [16]. Their lungs were severely atelectatic and development was retarded and remained in the early canalicular stage, accompanied by a reduction of *ENaC* mRNA levels in total lung RNA [16]. Several studies demonstrated the stimulation of *ENaC* subunit expression by GCs [17,18]. However, the sex difference in RDS persisted despite the routine use of antenatal GCs [19]. Furthermore, it is still controversial whether male and female preterm infants benefit equally from this therapeutic strategy [20,21,22]. We therefore aimed to analyze the effect of GCs on alveolar Na^+^ transport and channel expression in FDLE cells of both sexes to determine whether a sex-specific responsiveness to GCs exists. To this end, FDLE cells were cultured in serum-free medium (SF-Med), which eliminated the sex difference in Na^+^ transport and channel expression, suggesting that sex differences are induced by serum components like steroids.

## 2. Results

### 2.1. Glucocorticoid Receptor mRNA Expression in Sex-Specific FDLE Cells

Assuming a potential sex difference in GC responsiveness, we first determined whether the *GR* mRNA level was differentially expressed between male and female FDLE cells cultured in SS-Med. Indeed, *GR* mRNA expression was significantly higher in female FDLE cells (*p* < 0.05; Figure 1A), possibly rendering them more receptive to GCs. Higher *GR* mRNA expression in female FDLE cells was observed in cells cultured in SS-Med for 3 days (Figure 1A) as well as in freshly isolated fetal lung cells without prior cell culture (*p* < 0.01; Figure 1B). Serum deprivation using SF-Med abolished the sex difference in *GR* mRNA expression seen in SS-Med (Figure 1A). This suggests that fetal serum factors, present in SS-Med and in the fetal circulation itself, induce the differential *GR* mRNA expression observed between male and female cells. This led us to analyze the effect of androgens and female sex steroids on *GR* mRNA expression. For this analysis, SF-Med was used to exclude the influence of other steroids possibly present in the fetal serum. *GR* mRNA expression in male and female FDLE cells was not affected by androgens (Figure 1C). In addition, neither estrogen nor progesterone exerted a significant effect on *GR* mRNA expression in male and female FDLE cells (Figure 1D). These results demonstrate that under physiological conditions, *viz.* the presence of fetal serum, sex differences exist, but neither androgens nor female sex steroids affect *GR* mRNA expression.

### 2.2. Glucocorticoids Increase Na^+^ Transport and Channel Expression in the Absence of Serum

We addressed the effect of dexa on epithelial Na^+^ channels in male and female FDLE cells separately. Cells were cultured in SF-Med to exclude potential effects of serum steroids. Ussing chamber measurements showed that baseline *I*_SC_ (*I*_base_) of male cells was significantly increased by 10 nM and 100 nM dexa compared to unstimulated male control cells (*p* < 0.001; Figure 2A). The lowest concentration of dexa (1 nM) did not significantly increase *I*_base_ in male cells. The amiloride-sensitive *I*_SC_ (Δ*I*_amil_) was significantly increased by all tested dexa concentrations in male FDLE cells (*p* < 0.001; Figure 2A). Similarly, 1, 10 and 100 nM dexa significantly increased *I*_base_ and Δ*I*_amil_ in female FDLE cells (*p* < 0.05; *p* < 0.01; *p* < 0.001; Figure 2B). The results thus suggest that epithelial Na^+^ transport of male and female FDLE cells is equally induced by all tested GC concentrations.

To analyze whether another GC, betamethasone (beta), exerts sex-specific effects in comparison to dexa, we determined Na^+^ transport in response to two beta and one dexa concentration in SF-Med. The *I*_base_ of male cells was significantly increased by 100 nM and 500 nM beta, as well as 100 nM dexa compared to unstimulated male control cells (*p* < 0.01; *p* < 0.001; Figure 2C). The Δ*I*_amil_ was further significantly increased by all tested GC concentrations in male FDLE cells (*p* < 0.001; Figure 2C). Similarly, the GCs significantly increased *I*_base_ and Δ*I*_amil_ in female FDLE cells (*p* < 0.05, *p* < 0.01, *p* < 0.001; Figure 2D). FDLE cells of both sexes are therefore equally stimulated by the tested GCs.

The maximal amiloride-sensitive apical membrane permeability (*amil*_max_) was significantly increased by 1 nM dexa in male FDLE cells (*p* < 0.001; Figure 3A) and similarly in female FDLE cells (*p* < 0.001; Figure 3B). In another set of experiments, the apical membrane was permeabilized loading the cell interior and the Na,K-ATPase with Na^+^, which enables determination of the maximal ouabain-sensitive *I*_SC_ (*ouab*_max_). *Ouab*_max_ was not affected by dexa stimulation (1 nM) in both male (Figure 3C) and female FDLE cells (Figure 3D) compared to unstimulated control cells. In summary, dexa highly stimulates apical ENaC activity in both male and female FDLE cells cultured in SF-Med, whereas the basolateral Na,K-ATPase activity was not affected by GCs in both sexes.

Next, we analyzed the effect of different dexa concentrations in SF-Med on *ENaC* and *Na,K-ATPase* mRNA expression in male and female FDLE cells. In male cells, 10 nM and 100 nM dexa increased *α-ENaC* mRNA expression 10-fold and 14-fold (*p* < 0.001; Figure 4A). The *β-ENaC* and *γ-ENaC* subunit mRNA expression were increased 2-fold and 2.5-fold by 10 and 100 nM dexa, respectively (*p* < 0.001; Figure 4B). The *Na,K-ATPase* subunit *α_1_* was increased 1.5-fold and 1.7-fold, while subunit *β**_1_* was elevated 3.7-fold and 4.2-fold by 10 and 100 nM dexa in male cells (*p* < 0.01; *p* < 0.001; Figure 4B). The lowest dexa concentration of 1 nM elevated the *α-ENaC* mRNA expression 2.4-fold (*p* < 0.001; Figure 4A), but did not affect any other subunit in male FDLE cells (Figure 4B). Similar results were obtained for female FDLE cells, in which 10 nM and 100 nM dexa increased *α-ENaC* mRNA 9-fold and 11-fold (*p* < 0.001; Figure 4C). The *β-ENaC* and *γ-ENaC* subunit expression were increased 2-fold and 2.5-fold by 10 and 100 nM dexa, respectively (*p* < 0.001; Figure 4D). Furthermore, in females the *Na,K-ATPase* subunits were elevated 1.4-fold and 1.6-fold for *α_1_*, and 3.1-fold and 4.3-fold for *β**_1_* (*p* < 0.05; *p* < 0.01; *p* < 0.001; Figure 4D). In contrast to males, the lowest concentration of dexa (1 nM) not only increased female *α-ENaC* mRNA 2.5-fold (*p* < 0.001; Figure 4C), but also significantly elevated *β-ENaC* 1.2-fold, *Na,K-ATPase-α_1_* 1.4-fold and *-**β**_1_* 1.3-fold (*p* < 0.05; *p* < 0.01; Figure 4D). In summary, the only sex difference was observed with a low dexa concentration in females displaying elevated *β-ENaC* and *Na,K-ATPase* subunit expression, whereas these subunits were not responsive in male FDLE cells.

To determine whether inhibition of the GR would abolish the dexa-induced increase in Na^+^ transport, we used mifepristone, a highly affine GR antagonist, in SF-Med. In line with the previous measurements, dexa strongly elevated Δ*I*_amil_ in male and female FDLE cells (*p* < 0.001; Figure 5A,B). GR inhibition by mifepristone significantly reduced the dexa-evoked stimulation of Na^+^ transport (*p* < 0.05), thereby abolishing the increasing effect of dexa in both male and female FDLE cells. In accordance, mRNA expression of the Na^+^ transporters was elevated by dexa (*p* < 0.001; Figure 6A−D), which was significantly reduced by mifepristone in male and female FDLE cells (*p* < 0.001; Figure 6A−D). Thus, inhibition of the GR prevented the stimulating effect of dexa on Na^+^ transporter activity and mRNA expression in both sexes.

### 2.3. Effect of Glucocorticoids on Surfactant Protein Expression

We further addressed the response of surfactant protein mRNA expression to dexa stimulation in SF-Med. Dexa concentrations of 10 nM and 100 nM increased *SP-A*, *SP-B* and *SP-C* mRNA expression in both, male and female FDLE cells (*p* < 0.05; *p* < 0.01; *p* < 0.001; Figure 7A,B). However, 1 nM dexa increased *SP-C* mRNA expression only in females (*p* < 0.01), while in male cells 1 nM dexa did not significantly increase mRNA expression of surfactant proteins. Thus, surfactant protein mRNA expression is more responsive to low GC stimulation in female FDLE cells compared with male cells. In contrast to 1 nM dexa, the response to 10 nM dexa was stronger in male FDLE cells in which *SP-A*, *SP-B* and *SP-C* were increased 8.2-fold, 6.3-fold and 3.0-fold, respectively. In females 10 nM dexa increased *SP-A*, *SP-B* and *SP-C* mRNA only 4.1-fold, 3.1-fold and 1.6-fold, respectively, which was approximately 50% lower compared to the RFC in males. The response of *SP-A* to 100 nM dexa was also more pronounced in male cells with a 19.7-fold increase in *SP-A*, compared to the 12.8-fold increase in females. The response of *SP-B* and *SP-C* mRNA levels to 100 nM dexa was not different between male and female cells, with increases of approximately 8.4-fold and 2.7-fold for both sexes.

### 2.4. Effect of Glucocorticoids on Na^+^ Transport and Channel Expression in the Presence of Serum

The responsiveness to GCs was determined in the absence of serum, because fetal serum contains several growth factors and possibly steroids that could obscure the response to GCs. Serum deprivation demonstrated that the stimulation of Na^+^ transport by dexa is largely independent of sex. However, these results were obtained under unphysiological conditions, *viz.* the absence of serum. We therefore repeated the measurement of Na^+^ transport in Ussing chambers with male and female cells cultured with SS-Med and stimulated with dexa for 48 h. In control cells, a significant difference of *I*_base_ and Δ*I*_amil_ (*p* < 0.01; Figure 8A,B) was observed between male and female FDLE cells, with females exhibiting a higher Na^+^ transport. Surprisingly, dexa addition did not elevate *I*_base_ or Δ*I*_amil_ in male and female FDLE cells, but abolished the sex difference by decreasing female *I*_base_ and Δ*I*_amil_ to the male level (*p* < 0.01; Figure 8A,B). To determine whether inhibition of the GR would abolish the sex difference in Na^+^ transport observed in SS-Med, mifepristone was used. GR inhibition by mifepristone did not equalize Na^+^ transport in FDLE cells since *I*_base_ and Δ*I*_amil_ were still significantly higher in female FDLE cells compared with male cells (*p* < 0.05; *p* < 0.01; Figure 8C,D). These results therefore suggest that the response to GCs depends on the presence of serum, and that the sex difference in epithelial Na^+^ transport of male and female FDLE cells is not mediated via the GR.

In contrast to the Na^+^ transport activity, the mRNA expression was strongly elevated by dexa even in the presence of serum (*p* < 0.001; Figure 9A−E). Both, male and female FDLE cells showed increased *ENaC* and *Na,K-ATPase* subunit mRNA expression in response to dexa, which was reduced by mifepristone (*p* < 0.01, *p* < 0.001; Figure 9A−E). Although in the presence of serum, mifepristone was rarely able to block the stimulating effect of dexa completely. Comparing the dexa-induced response of male and female FDLE cells, dexa abolished the sex difference observed in control cells by equally raising *ENaC* mRNA expression in both sexes (*p* < 0.05; *p* < 0.01; Figure 9C), with the exception of *α-ENaC*. In male FDLE cells, *α-ENaC* was increased 14.2-fold by dexa and in female cells 9.6-fold, resulting in a significantly lower mRNA expression in females (*p* < 0.05). In accordance to the Ussing chamber results, dexa abolished the sex difference observed in the presence of serum, however; it did so by stimulating the Na^+^ transporter mRNA expression in both sexes, while the Na^+^ transport activity was equalized by reducing the activity in females.

## 3. Discussion

Male sex represents a major risk factor for developing respiratory complications, especially in preterm infants [3,23,24]. It is still largely unknown, why male infants are more susceptible to develop respiratory distress [5]. GCs play a key role in lung development by increasing the mRNA expression and functional activity of ENaC [25,26] and thereby elevating Na^+^ absorption and AFC [18]. However, the use of antenatal GCs and postnatal surfactant, although improving overall survival, has not narrowed the gap in RDS-related mortality between boys and girls [3,4,19,27]. A sex-specific effect of antenatal GC therapy has been proposed, although clinical evidence is controversial [20,21,22,28] and mechanistic studies have not been conducted on this issue. To our knowledge, this is the first study addressing sex-specific effects of GCs on alveolar epithelial Na^+^ transport in fetal alveolar cells. Our results suggest that GCs increase Na^+^ transport in both sexes and differences are at best marginal. Furthermore, GR inhibition did not equalize the sex differences in Na^+^ transport. We therefore conclude that sex differences observed for respiratory complications associated with preterm birth are unlikely due to differential responsiveness of Na^+^ transport to antenatal GC therapy.

Liggins and Howie’s publication in 1972 first demonstrated beneficial effects of antenatal GC administration on RDS and perinatal morbidity [29]. Antenatal GC use became widely accepted in the early 1990s, particularly after the National Institutes of Health Consensus Conference in 1994 Consensus Development Conference [30]. In a double-blind, randomized trials of antenatal GC therapy, the Collaborative Group on Antenatal Steroid Therapy observed a definitive benefit in girls, while no benefits were evident in boys [31]. In contrast, Liggins and Howie had previously observed a definitive benefit in boys but no significant benefit in girls [29]. These two large studies showed contrasting results with regard to the role of neonatal sex in response to antenatal GC therapy. A systematic literature review and meta-analysis of randomized controlled trials to evaluate the role of fetal sex in response to antenatal GCs detected no influence of neonatal sex on reduction of RDS or other neonatal outcomes [22]. In summary, a sex-specific effect of GCs remains inconclusive, and mechanistic studies addressing important functional targets of antenatal GC therapy, like alveolar Na^+^ transport, might thus advance our current knowledge of this topic.

Addressing a potential sex-specific expression of the GR, we found that female FDLE cells indeed had a higher *GR* mRNA expression compared to male cells when cultured in SS-Med. In contrast, culturing the cells in SF-Med abolished this sex difference. On the other hand, direct RNA isolation without prior cell culture confirmed the higher *GR* mRNA expression in females observed in the presence of fetal serum. We next tested whether male or female sex steroids affected alveolar *GR* mRNA expression and found that androgens did not affect *GR* mRNA expression in male or female FDLE cells. This not surprising as we have previously shown that androgen receptor-specific mRNA expression was not detected in male or female FDLE cells [11]. It is thus unlikely that the presence of male sex steroids lowers the *GR* mRNA levels. In contrast, a previous study demonstrated a negative effect of androgens on *GR* mRNA and protein levels, but in whole lung tissue [32]. Female sex steroids had no influence on fetal pulmonary *GR* mRNA expression, while the mRNA expression of the estrogen receptor-β and the progesterone receptor has been observed in FDLE cells [33]. In contrast to our results, the total number of lung GR and the measured binding affinity did not significantly differ between male and female sheep fetuses [34]. The different outcomes might be due to differences in species or gestational age or the fact that we analyzed *GR* mRNA expression in contrast to the GR protein content measured in fetal sheep.

Several studies demonstrated a stimulation of *ENaC* and *Na,K-ATPase* mRNA expression and activity in response to GCs [25,35,36,37,38]; however, sex-specific studies have not been conducted yet. Since cord serum levels of GCs after antenatal steroid treatment were shown to be similar in infants of both sexes, and differences between GC transfer or metabolism were not observed [28], we analyzed the responsiveness of male and female FDLE cells to different GC concentrations in Ussing chambers. Concentrations as low as 1 nM dexa increased ENaC activity in both male and female cells, demonstrating a similar stimulation of Na^+^ transport in both sexes. In addition, dexa as well as beta increased Na^+^ transport in male and female cells equally. The maximal ENaC activity was increased in both male and female FDLE cells by 1 nM dexa, whereas the maximal Na,K-ATPase activity was not affected by 1 nM dexa in FDLE cells of both sexes. These measurements demonstrate that no sex difference in responsiveness of Na^+^ transport to GCs exists in fetal alveolar cells. In addition, *ENaC* and *Na,K-ATPase* subunit mRNA expression was analyzed with different GC concentrations. The *α-ENaC* subunit mRNA expression was highly responsive to GCs in both sexes with 1 nM dexa increasing it by more than twofold. Other studies reported that *α-ENaC* is expressed at the end of gestation [39,40], whereas the *β-* and *γ*-subunit are predominantly expressed after birth [41,42]. Because FDLE cells are fetal cells, a strong effect on *α-ENaC* might be explicable. Higher dexa concentrations (10 and 100 nM) elevated mRNA expression of *β-* and *γ-ENaC* as well as *Na,K-ATPase-α_1_* and *-β_1_* in both sexes. The only sex difference was observed with 1 nM dexa which increased *β- ENaC* as well as *Na,K-ATPase-α_1_* and *-β_1_* only in female FDLE cells. However, this increase of *Na,K-ATPase* mRNA expression in females was not sufficient to elevate the maximal Na,K-ATPase activity as detailed above. In conclusion, a marginal sex difference was observed for some *ENaC* and *Na,K-ATPase* subunits being more responsive to GCs in females, which, however lacked any functional consequence in channel activity as shown by Ussing chamber measurements. These results led us to question an involvement of GCs in the sex difference observed for Na^+^ transport.

Finally, the effects of different GC concentrations on mRNA expression of *SP-A*, *SP-B* and *SP-C* were analyzed in male and female FDLE cells. *SP-C* mRNA expression was more responsive to low dexa concentrations in female cells. In contrast, 1 nM did not significantly increase *SP-C* mRNA expression in male cells. Higher GC concentrations increased *SP-A*, *SP-B*, and *SP-C* mRNA expression in both male and female cells, but male cells demonstrated higher increases compared with females. These results thus suggest that fetal alveolar cell exhibit a sex difference in surfactant protein mRNA expression in response to GCs, with females being more responsive. But this difference was only observed with the lowest concentration used, and further elevating the dexa concentration achieved an equal or even higher effect in male cells. Whether the differential responsiveness to GCs observed for surfactant protein mRNA expression is an explanation for the “male disadvantage” or why male preterm infants require more doses of surfactant than female infants [43], needs to be determined in future studies. According to a clinical study, standard antenatal steroid treatment resulted in 6–10 µg/dL (166–276 nM) cortisol equivalents in cord serum of treated premature infants [28]. The used concentration of 1 nM dexa equates to 30 nM cortisol equivalents, which is approximately 5–9 times lower than the serum concentration achieved in the infants after treatment. We therefore suggest that the difference observed with 1 nM dexa is not relevant to the clinical situation.

Of note, the experiments testing the response to GCs were conducted in cells cultured in SF-Med, to exclude an effect of other steroids possibly present in the fetal serum. Furthermore, prior studies analyzing the effect of GCs, have always been performed in SF-Med supplemented with dexa [18,35,36,38], in accordance with our experiments. However, culturing FDLE cells in SF-Med is not a physiological condition and the sex difference between male and female cells was only observed in the presence of fetal serum, either by the addition of cell culture serum or in the fetus itself. We therefore conducted additional measurements in which the cells were continuously grown in SS-Med and stimulated with GCs. To our surprise, the addition of GCs in the presence of serum did not increase Na^+^ transport in male or female FDLE cells. Moreover, the addition of GCs abolished the otherwise observed sex difference in the presence of serum by reducing the Na^+^ transport of females to the male level. Assuming that the presence of fetal serum represents a more physiological condition for the cells, a stimulating effect of GCs on epithelial Na^+^ transport cannot be confirmed. To our knowledge this is the first study analyzing the effect of GCs on Na^+^ transport in FDLE cell culture in the presence of serum and the effects observed in SF-Med by other groups might not represent the actual situation.

To exclude a contribution of the GR to the sex difference in Na^+^ transport observed in the presence of serum, we blocked the GR with mifepristone. The addition of mifepristone to SS-Med had no impact on the sex difference in Na^+^ transport, since female FDLE cells still showed a significantly higher ENaC activity. Although, females possess a higher GR mRNA expression as detailed above, these results suggest that the GR activity is not causal for the sex difference in Na^+^ transport.

Finally, mRNA expression of the Na^+^ transporters was measured in FDLE cells cultured in SS-Med and stimulated with GCs. Interestingly, dexa significantly increased the mRNA expression of all *ENaC* and *Na,K-ATPase* subunits in both male and female FDLE cells. This is in strong contrast to the results obtained in Ussing chamber measurements, since Na^+^ transport was not stimulated by dexa in the presence of serum, suggesting that the increased mRNA expression observed does not result in an increased ENaC activity. Nevertheless, the addition of dexa abolished the sex difference in *ENaC* mRNA expression by increasing the expression in both sexes to equal levels, with the exception of *α-ENaC* that was more stimulated in males.

Taken together, the results do not support a sex-specific effect of GCs on epithelial Na^+^ transport. Neither in the presence or absence of serum, a pronounced difference between male and female cells was observed. Furthermore, the effects of GCs greatly depend on the cellular environment, which must be considered in future studies.

We did not perform analyses of protein expression, which is a limitation of our study. Protein expression measurements would for instance elucidate whether the higher *GR* mRNA expression in females also results in a higher GR protein content. However, since the differential *GR* mRNA expression had no functional impact on the response to GCs, we assumed that analysis of protein content was not essential to answer our research question. The same accounts for the *ENaC* and *Na,K-ATPase* subunit mRNA expression and their marginal differences in response to GCs, since these differences lacked any functional consequence.

Furthermore, the results obtained in our study are only applicable to the developmental age of our cells. A study in fetal rabbits demonstrated sex differences in fetal pulmonary surfactant production only during a limited developmental time frame, since very immature as well as mature fetuses exhibited no sex difference [44]. The cells used in our study were isolated in the late canalicular to early saccular stage, 24–48 h prior to birth, which overlaps with the human preterm infant prior to the 36th week of gestation. However, human preterm infants are born between the 24th to 36th week of gestation, which is an extensive period, but male sex has been repeatedly shown to be an independent risk factor for respiratory distress in infants of 24–34th gestational weeks [3,43]. Therefore, the risk factor of male sex is present throughout the saccular stage and not dependent on a certain developmental age. Still we cannot exclude that a sex-related responsiveness towards antenatal GCs might vary depending on the age of the infants. However, our cells exhibit sex differences in Na^+^ transport, also independent of cell culture, correlating with lung fluid content [8] as a physiological outcome and we therefore assume that the developmental age of our cells is appropriate to determine the responsiveness to GCs.

## 4. Materials and Methods

### 4.1. Cell Isolation and Culture

The institutional review board (IRB: Landesdirektion Leipzig, permit number: T23/15) approved all experimental procedures (1 February 2016). Sprague-Dawley rats were housed at the Medical Experimental Center (MEZ) of Leipzig University. Rats were kept in rooms with a 12 h light-dark cycle, constant temperature (22 °C) and humidity (55%). Food and water were supplied *ad libitum.* Pregnant rats were anesthetized by CO_2_ inhalation and euthanized by Pentobarbital injection at gestational day E20-21 (term E = 22). Separation of fetuses by sex was done through visual determination [45,46], as described before [8].

To model preterm ATII cells, FDLE cells were isolated from fetal lungs as described previously [18,47]. Male and female fetuses, derived from the same litter in equal numbers, were prepared separately. Cell isolation was done in parallel. After mechanical dissociation the cell suspension was enzymatically digested by incubation in HBSS (Life Technologies, Darmstadt, Germany) with 0.125 % trypsin (Life Technologies) and 0.4 mg/mL DNAse (CellSystems, Troisdorf, Germany) for 10 min at 37 °C and MEM containing 0.1% collagenase (CellSystems) and DNAse for 15 min at 37 °C. FDLE cells were seeded on permeable Snapwell inserts (Costar, # 3407, surface area 1.1 cm^2^, Corning Inc., Corning, NY, USA) at a density of 10^6^ cells per insert for the electrophysiological measurements. For RNA isolation, cells were seeded on Transwell inserts (Costar, # 3412, surface area 4.6 cm^2^) at a density of 2 × 10^6^ cells per insert. The serum-supplemented medium (SS-Med) consisted of MEM with 10% FBS (Biochrom, Berlin, Germany), glutamine (2 mM, Life Technologies) and antibiotic-antimycotic (Life Technologies). After 24 h cell culture media was either changed to SF-Med (Cellgro, Mediatech, Herndon, VA, USA) for experiments testing the isolated effect of GCs, or continuously grown in SS-Med. FDLE cells were stimulated with dexamethasone (dexa, # D4902, Sigma-Aldrich, Taufkirchen, Germany) or betamethasone (beta, Celestan^®^; MSD Sharp & Dohm GmbH, Haar, Germany). The antagonist mifepristone (10 µM, # M8046, Sigma-Aldrich) was used to block the GR and thereby the dexa-evoked response. Hormones and receptor antagonists were dissolved in DMSO. To exclude solvent influences on the evoked responses, control cells were treated with same amount of the respective solvent.

### 4.2. Ussing Chamber Measurements

Ussing chamber measurements were performed five days after cell isolation, as previously reported [8]. Only monolayers with a transepithelial resistance (*R*_te_) exceeding 300 Ω·cm^2^ were included in the analyses. Electrophysiological solutions consisted of: 145 mM Na^+^, 5 mM K^+^, 1.2 mM Ca^2+^, 1.2 mM Mg^2+^, 125 mM Cl^−^, 25 mM HCO_3_^−^, 3.3 mM H_2_PO_4_^−^ and 0.8 mM HPO_4_^2−^ (pH 7.4). While 10 mM glucose was used for the basolateral solution, 10 mM mannitol was used in the apical solution. During measurements, the solutions were continuously bubbled with carbogen (5% CO_2_ and 95% O_2_). Equivalent short-circuit currents (*I*_SC_) were determined every 20 s by measuring transepithelial voltage (*V*_te_) and *R*_te_ with a transepithelial current clamp (Physiologic instruments, San Diego, CA) and calculating the quotient *I*_SC_ = *V*_te_/*R*_te_. After the *I*_SC_ reached a stable plateau (*I*_base_), amiloride (10 µM, # A7410, Sigma-Aldrich) was applied to the apical chamber to assess the amiloride-sensitive Δ*I*_SC_ (Δ*I*_amil_). The current reduction induced by amiloride (Δ*I*_amil_) was used as a measure of ENaC activity. The amiloride-sensitive apical membrane permeability (*amil*_max_) was measured in the presence of a 145:5 apical to basolateral Na^+^ gradient and a permeabilized basolateral membrane using 100 µM amphotericin B (a pore forming antibiotic, 100 µM, # A-4888, Sigma-Aldrich). To determine maximal Na,K-ATPase capacity, the apical membrane was permeabilized with 10 µM amphotericin B. The *I*_SC_ was measured every 5 s with a transepithelial voltage clamp. When the *I*_SC_ had risen to its maximum value, ouabain (1 mM, # O3125, Sigma-Aldrich) was added to the basolateral compartment and the ouabain-sensitive component of the amphotericin-induced maximal *I*_SC_ (*ouab*_max_) was calculated. Amiloride and ouabain stock solutions were prepared in water.

### 4.3. mRNA Expression Analyses

RNA isolation was done five days after cell isolation using the Purelink RNA Mini Kit (Life Technologies) and DNAse I (Life Technologies) according to the manufacturer’s instructions. Reverse transcription was carried out with pre-annealing of 1 µg RNA with Oligo(dT)_18_ primers (Fisher Scientific GmbH, Schwerte, Germany) and subsequent addition of Superscript III (Life Technologies) for 1 h at 55 °C and 15 min at 75 °C. Real-time quantitative PCR (RT-qPCR) was done in the CFX 96 Real-Time system (Bio-Rad, Munich, Germany) using the SYBR Select Master Mix (Fisher Scientific GmbH) and gene-specific primers. Absolute quantification was done with a serial dilution of target-specific plasmid DNA as internal standard curve. The resulting molecule concentrations were normalized to a reference gene encoding for the *60S ribosomal protein L13a* (*Rpl13a*) [48]. Constant expression of *Rpl13a* was confirmed against other common reference genes. The mRNA levels were calculated with the relative standard curve method and expressed as relative fold change (RFC) of the respective control. Melting curves and gel electrophoresis of PCR products were routinely performed to control the specificity of the PCR reaction.

### 4.4. Statistical Analysis

Statistical analysis was performed with GraphPad Prism software (GraphPad Software, La Jolla, CA, USA). Differences between two groups were analyzed with the unpaired *T*-test. Otherwise, significant differences were determined by one-way or two-way ANOVA. A probability of *p* < 0.05 was considered significant for all statistical analyses.

## 5. Conclusions

In summary, female and male alveolar cells exhibit no sex difference in responsiveness to GCs with regard to Na^+^ transport, and GR activity does not contribute to the higher Na^+^ transport in females. At least for Na^+^ transport and the related AFC, a sex-specific effect of antenatal GC therapy cannot be demonstrated.

## Figures and Tables

**Figure 1 ijms-21-00057-f001:**
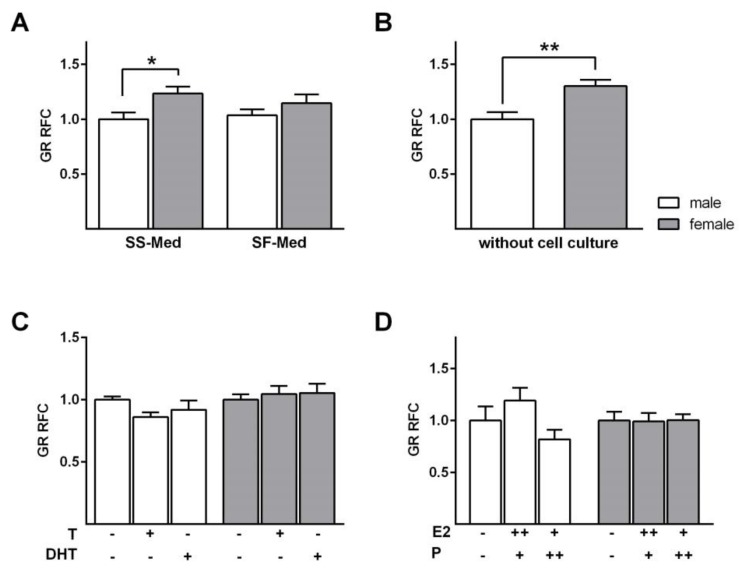
*GR* mRNA expression in sex-specific FDLE cells. FDLE cells were isolated from fetal rats. Data bars represent the relative fold change (RFC) + SEM. (**A**) FDLE cells were cultured in SS-Med or SF-Med. *GR* mRNA expression was significantly higher in females cultured in SS-Med compared to equally treated male FDLE cells (*n* = 8, * *p* < 0.05, *T*-test). No significant sex difference was observed for *GR* mRNA expression in SF-Med. (**B**) RNA directly isolated from FDLE cells without prior cell culture showed significantly higher *GR* mRNA expression in females (*n* = 6, ** *p* < 0.01, *T*-test). (**C**) Testosterone (T) or dihydrotestosterone (DHT), in SF-Med, had no effect on *GR* mRNA expression in male and female cells (*n* = 8). (**D**) Progesterone (P) and estradiol (E2), in SF-Med, did not alter *GR* mRNA expression of male and female FDLE cells (*n* = 6). (□) male, (■) female.

**Figure 2 ijms-21-00057-f002:**
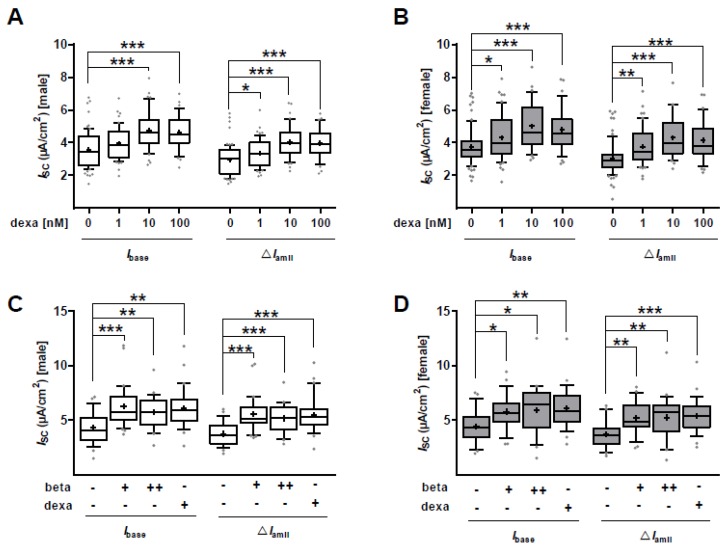
GCs increase Na^+^ transport in sex-specific FDLE cells. FDLE cells were cultured in SF-Med and stimulated with GCs for 48 h. Data are displayed as box and whiskers with the 10−90 percentile, mean (+) and median (horizontal line). In male (**A**) and female (**B**) FDLE cells, dexa significantly increased *I*_base_ and Δ*I*_amil_ (*n* = 40–54; * *p* < 0.05; ** *p* < 0.01; *** *p* < 0.001; one-way ANOVA). Moreover, in male (**C**) and female (**D**) FDLE cells, 100 nM dexa (+), 100 nM beta (+) and 500 nM beta (++) significantly increased *I*_base_ and Δ*I*_amil_ (*n* = 24–30; * *p* < 0.05; ** *p* < 0.01; *** *p* < 0.001; one-way ANOVA). (□) male, (■) female.

**Figure 3 ijms-21-00057-f003:**
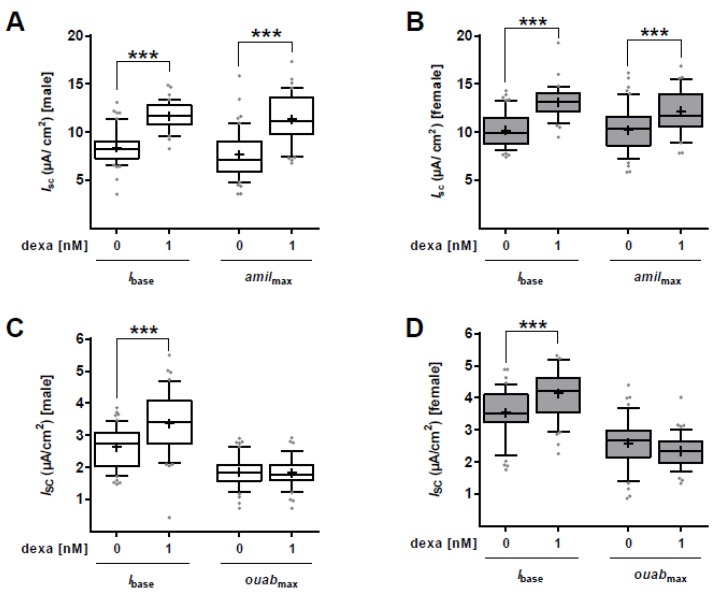
GCs increase *amil*_max_, but not *ouab*_max_ in sex-specific FDLE cells. FDLE cells were cultured in SF-Med and stimulated with dexa (1 nM) for 48 h. Data are displayed as box and whiskers with the 10-90 percentile, mean (+) and median (horizontal line). Dexa significantly increased *amil*_max_ in male (**A**) (*n* = 40–45; *** *p* < 0.001, *T*-test) and female (**B**) (*n* = 35–42; *** *p* < 0.001, *T*-test) FDLE cells. *Ouab*_max_ was not affected by dexa in male (**C**) (*n* = 48–52) and female (**D**) (*n* = 44–45) FDLE cells. (□) male, (■) female.

**Figure 4 ijms-21-00057-f004:**
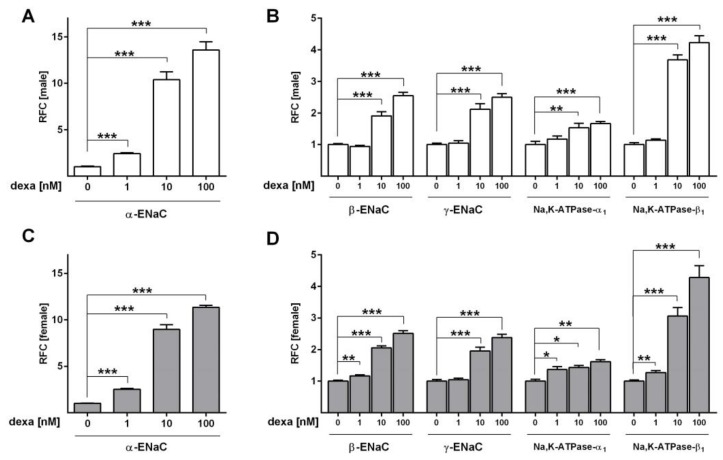
GCs elevate Na^+^ channel subunit mRNA expression in sex-specific FDLE cells. FDLE cells were cultured in SF-Med and stimulated with dexa for 48 h. Data bars represent the relative fold change (RFC) + SEM. In male (**A**) and female (**C**) FDLE cells, 1-100 nM dexa significantly increased *α-ENaC* mRNA expression (*n* = 6; *** *p* < 0.001; *T*-test). Moreover, dexa (10-100 nM) significantly increased *β-ENaC*, *γ-ENaC*, *Na,K-ATPase-α_1_* and *Na,K-ATPase-β_1_* mRNA expression in male (**B**) and female (**D**) FDLE cells (*n* = 6; * *p* < 0.05; ** *p* < 0.01; *** *p* < 0.001; two-way ANOVA). In addition, 1 nM dexa further increased *β-ENaC*, *Na,K-ATPase-α_1_* and *Na,K-ATPase-β_1_* mRNA expression in female cells (*n* = 6; * *p* < 0.05, ** *p* < 0.01, *T*-test). (□) male, (■) female.

**Figure 5 ijms-21-00057-f005:**
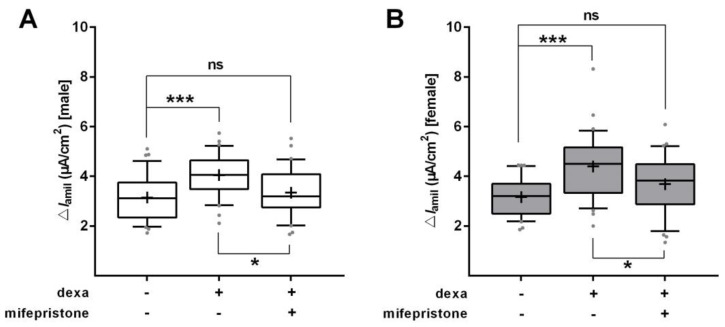
GR inhibition prevents the GC-evoked increase of Na^+^ transport in sex-specific FDLE **cells.** FDLE cells were cultured in SF-Med and stimulated with dexa (100 nM) with or without mifepristone for 48 h. Data are displayed as box and whiskers with the 10–90 percentile, mean (+) and median (horizontal line). In male (**A**) and female (**B**) FDLE cells, dexa significantly increased Δ*I*_amil_, while mifepristone significantly reduced the dexa-induced response (*n* = 40–47; * *p* < 0.05; *** *p* < 0.001; two-way ANOVA). (□) male, (■) female.

**Figure 6 ijms-21-00057-f006:**
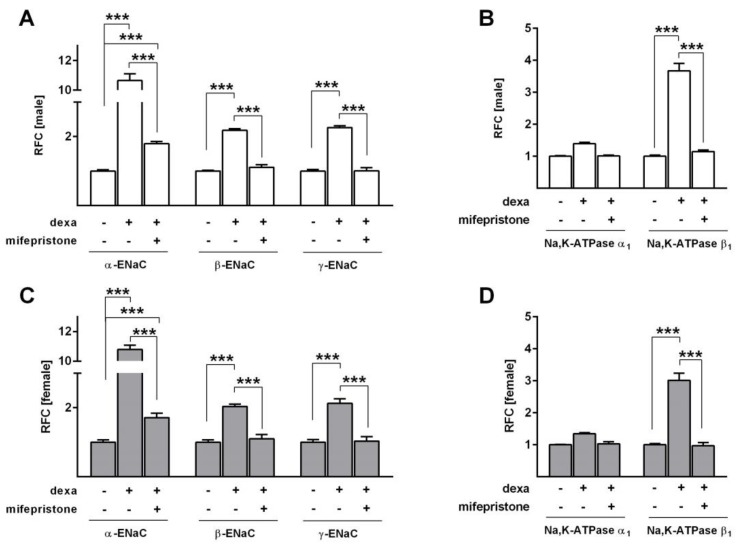
GR inhibition prevents the GC-evoked increase of Na^+^ channel subunit expression in sex-specific FDLE cells. FDLE cells were cultured in SF-Med and stimulated with dexa (100 nM) with or without mifepristone for 48 h. Data bars represent the relative fold change (RFC) + SEM. In male (**A**) and female (**C**) FDLE cells, dexa significantly increased *α-ENaC*, *β-ENaC* and *γ-ENaC* mRNA expression (*n* = 6; *** *p* < 0.001), while mifepristone significantly reduced or completely prevented the dexa-induced response (*n* = 6; *** *p* < 0.001; two-way ANOVA). Furthermore, dexa significantly increased *Na,K-ATPase-β_1_* mRNA expression in male (**B**) and female (**D**) cells, which was reduced or prevented by mifepristone (*n* = 6; *** *p* < 0.001; two-way ANOVA). (□) male, (■) female.

**Figure 7 ijms-21-00057-f007:**
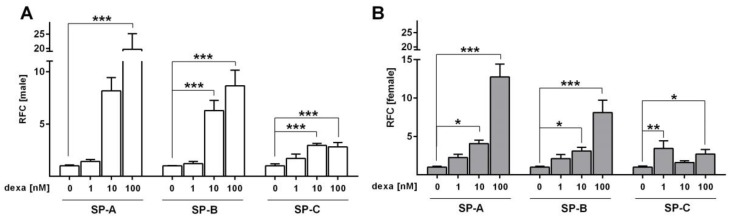
GCs elevate surfactant protein expression in sex-specific FDLE cells. FDLE cells were cultured in SF-Med and stimulated with dexa for 48 h. Data bars represent the relative fold change (RFC) + SEM. (**A**) In male FDLE cells, 10–100 nM dexa significantly increased *SP-A*, *SP-B* and *SP-C* mRNA expression (*n* = 6; *** *p* < 0.001; two-way ANOVA). (**B**) In female FDLE cells, 1−100 nM dexa significantly increased *SP-A*, *SP-B* and *SP-C* mRNA expression (*n* = 6; * *p* < 0.05; ** *p* < 0.01; *** *p* < 0.001; two-way ANOVA). (□) male, (■) female.

**Figure 8 ijms-21-00057-f008:**
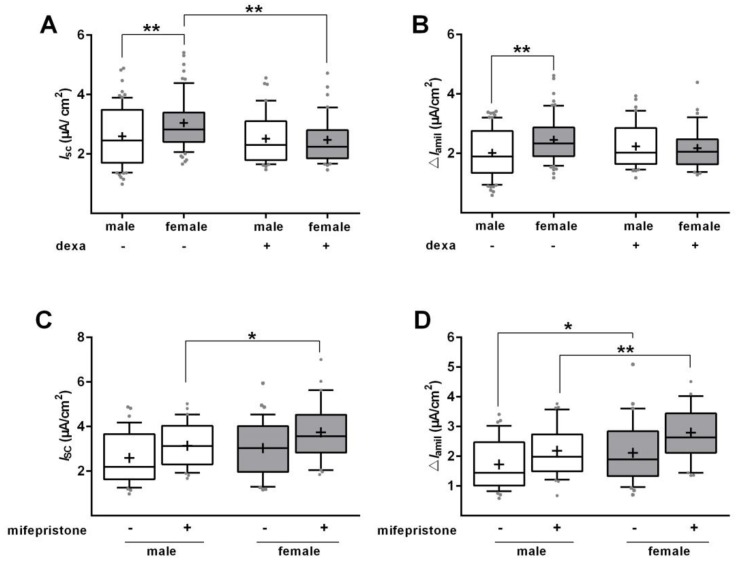
Effects of GCs and GR inhibition on Na^+^ transport in the presence of serum. Data are displayed as box and whiskers with the 10-90 percentile, mean (+) and median (horizontal line). (**A**,**B**) FDLE cells were cultured in SS-Med and stimulated with dexa (100 nM) for 48 h. Sex-specific differences in *I*_base_ (**A**) and Δ*I*_amil_ (**B**) were observed between male and female FDLE cells. The addition of dexa did not stimulate Na^+^ transport in SS-Med cultured cells, but abolished the sex differences by reducing the Na^+^ transport in female FDLE cells to the male level (*n* = 38–40; ** *p* < 0.01, two-way ANOVA). (**C**,**D**) FDLE cells were cultured in SS-Med and mifepristone for 48 h. (**C**) *I*_base_ was significantly higher in female compared to male cells in the presence of mifepristone. (**D**) Mifepristone did not abolish the sex difference in Δ*I*_amil_, which was still significantly higher in female compared to male cells (*n* = 30–36; * *p* < 0.05; ** *p* < 0.01; two-way ANOVA). (□) male, (■) female.

**Figure 9 ijms-21-00057-f009:**
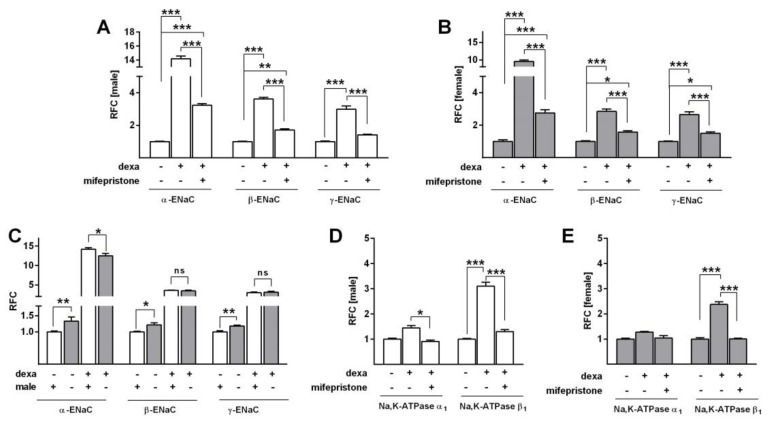
Effects of GCs and GR inhibition on Na^+^ channel subunit expression in the presence of serum. FDLE cells were cultured in SS-Med and stimulated with dexa (100 nM) with or without mifepristone for 48 h. Data bars represent the relative fold change (RFC) + SEM. In male (**A**) and female (**B**) FDLE cells, dexa significantly increased *α-ENaC*, *β-ENaC* and *γ-ENaC* mRNA expression (*n* = 6; *** *p* < 0.001). Mifepristone significantly reduced the dexa-induced response (*n* = 6; *** *p* < 0.001), although the mRNA expression remained in part significantly higher compared with controls (** *p* < 0.01; *** *p* < 0.001; two-way ANOVA). (**C**) Comparing *α-ENaC*, *β-ENaC* and *γ-ENaC* mRNA expression of male and female FDLE cells showed that only control cells, cultured in SS-Med, demonstrated sex differences (*n* = 6; * *p* < 0.05; ** *p* < 0.01; *T*-test), while these were abolished by the addition of dexa. Dexa raises the ENaC subunit mRNA expression of both sexes to similar levels, with the exception of *α-ENaC*, which was significantly higher expressed in males after the addition of dexa (*n* = 6; * *p* < 0.05, *T*-test). Furthermore, dexa significantly increased *Na,K-ATPase-β_1_* mRNA expression in male (**D**) and female (**E**) cells, which was reduced or prevented by mifepristone (*n* = 6; * *p* < 0.05; *** *p* < 0.001; two-way ANOVA). (□) male, (■) female.

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
