# Peer review of "Glucocorticoids Equally Stimulate Epithelial Na+ Transport in Male and Female Fetal Alveolar Cells"

_ijms, 2019, doi:10.3390/ijms21010057_

Round 1

Reviewer 1 Report

The authors have addressed sex differences between preterm newborns with special emphasis on the effect of glucocorticoids on epithelial sodium transport. The experiments were performed on "fetal distal lung epithelial cells"(FDLE cells) which were analysed/subjected to experiments five days after cell isolation. The introduction as well as the designed experiments are well described and planned. The results are well documented and appear sound, the discussion is a bit long but well-balanced.

However, I have some basic concerns that limit my enthusiasm about this manuscript:

Although I appreciate the amount of work put into the experiments, the manuscript only contains in vitro data. No in vivo experiments were performed to support the final conclusion of the manuscript. There are no data documenting the cells in culture. It would be important to know the composition of the "cellular cocktail" after isolation and at the time of the experiments. For example, AE2 cells tend to differentiate into AE1 cells in culture. Because of that I recommmend electron microscopy of the cultured cells after isolation and at day 5 after isolation. As the authors correctly point out in the discussion the developmental stage of the fetuses is extremely critical to the overall value of the study. Surprisingly, although the authors are aware of this, they have not attempted to provide suitable evidence that the time point of cell isolation really mimics the important critical developmental stage in humans. Finally, the rats from which the cells were isolated are surely between canalicular and saccular stage but it should be kept in mind that rats are physiologically born in the saccular stage.

These concerns make me feel, unfortunately, that the study design does not allow drawing the conclusion that there is no sex difference “in response to GCs with regard to Na+ transport” as stated in the abstract.

Author Response

We appreciate the critical revision of our study and want to respond to the concerns raised by the reviewer. Fetal distal lung epithelial cells are a well-established model of fetal ion transport and surfactant biosynthesis. Several groups have previously characterized the cellular composition of this culture, e.g. with electron microscopy or cell-specific antibodies [1,2]. Purity, based on reactivity with ATII cell-specific serum, was reported to be app. 91% for cells isolated from fetal rat lungs [1]. Removal of fibroblasts is achieved by differential adherence. For gene-expression-profiling, these primary cultures might not be pure enough, but they are well suited for studying transepithelial ion transport. Furthermore, these studies also showed that there was no significant difference in the percentage of newly synthesized phosphatidylcholine between the initially isolated ATII cells and those assayed after 7 days in culture [2]. This suggests that transdifferentiation of ATII to ATI cells is not very pronounced in FDLE culture. Support comes from another study demonstrating that the cellular structure was maintained for up to 8 days with the persistence of lamellar inclusion bodies [3]. In one of our previous studies, we also determined the mRNA expression of the Na,K-ATPase-α2 subunit, expressed by ATI-like cells, in our FDLE cells [4]. In freshly isolated cells and after 4 days in culture FDLE cells demonstrated a low but detectable expression of the α2 subunit, which however was about 1.000 times lower than α1-subunit levels. Although we do not think that a minor transdifferentiation of ATII to ATI cells would be problematic, as long as the transepithelial resistance is sufficient for Ussing chamber analysis, we think that the contribution of ATI cells is negligible within the period of our experiments. 

Although disease models will never mimic all aspects of the original pathology, we believe that the sex-specific FDLE cells are an appropriate model to study sex differences in Na+ transport and hence regulating factors. As the reviewer points out rodents are born at the saccular stage of lung development, however they do not develop respiratory distress in contrast to human preterm infants in whom RDS is frequently observed. We therefore believe that newborn rodents do not resemble the immaturity that lead to the respiratory insufficiency. However, rodents born prematurely, 24-48 h prior to term birth, exhibit respiratory distress and need to be rescued by oxygen supplementation. Therefore, from a developing perspective, the time point we use for our cell isolation is appropriate because of the shared lung immaturity. Furthermore, we observed sex differences in Na+ transport in our FDLE cells and sex differences in respiratory susceptibility have been described throughout the saccular stage in human preterm infants (24-34th weeks of gestational age). We added a paragraph about this in our manuscript. Thus, although the saccular stage in humans is an extensive period of app. 12 weeks, in contrast to rats with only app. 6 days, the sex differences in general persist throughout this stage. We therefore believe that the time we observe sex differences in our model is also the right time to study a potential regulation of this process by steroids. We also confirmed the lower Na+ transport in male FDLE cells with measurements of lung wet-to-dry ratio in fetal rats of the same age, representing lung fluid content. As expected, a lower Na+ transport in male rats led to a higher lung fluid content, shown in our previous study [4]. We were thereby able to correlate the results of the Ussing chambers with the physiological lung fluid content ex vivo. The reviewer further points out that we did not perform in vivo measurements, but measuring Na+ transport in vivo is not feasible. Only nasal potential measurements are sometimes used as a surrogate for transepithelial Na+ transport analysis, although this technique is error-prone and not feasible in prematurely born rodent pups that need immediate intensive care to survive. We therefore think that currently there is no reasonable approach available to study this process at this developmental age in vivo.

However, we adjusted our conclusion in the abstract to fit the experimental data to “there is no sex difference in FDLE cells in response to GCs with regard to Na+ transport”. We believe that we presented sufficient data to draw this conclusion.

Batenburg, J.J.; Otto-Verberne, C.J.; Have-Opbroek, A.A. ten; Klazinga, W. Isolation of alveolar type II cells from fetal rat lung by differential adherence in monolayer culture. Biochim. Biophys. Acta 1988, 960, 441–453, doi:10.1016/0005-2760(88)90053-7.

Post, M.; Torday, J.S.; Smith, B.T. Alveolar type II cells isolated from fetal rat lung organotypic cultures synthesize and secrete surfactant-associated phospholipids and respond to fibroblast-pneumonocyte factor. Exp Lung Res 1984, 7, 53–65, doi:10.3109/01902148409087908.

Fraslon, C.; Rolland, G.; Bourbon, J.R.; Rieutort, M.; Valenza, C. Culture of fetal alveolar epithelial type II cells in serum-free medium. In Vitro Cell. Dev. Biol 1991, 27A, 843–852, doi:10.1007/bf02630986.

Kaltofen, T.; Haase, M.; Thome, U.H.; Laube, M. Male Sex is Associated with a Reduced Alveolar Epithelial Sodium Transport. PLoS ONE 2015, 10, e0136178, doi:10.1371/journal.pone.0136178.

Reviewer 2 Report

The work offers interesting ideas for reflection and presents very interesting results.  I propose some minor revisions or suggestions  :

statistical analysis: How was the distribution assessed? due to the presence of different groups (sexes) and treatments, why was the most suitable 2-way anova not used? the authors should justify the use of the paired t-test and eventually add the correction for multiple comparisons (Bonferroni correction or false discovery rate)

Line 152-153: "These results demonstrate that under physiological conditions, viz. the presence of fetal serum, sex differences exist, but neither androgens nor female sex steroids strongly affect GR mRNA expression".  Do FDLE express (estrogen receptors) ERs and androgen receptor (AR)? Is their expression conditioned by fetal serum in the medium? Is their expression sex-dependent? May the lack of effect of sex steroids be due to the lack of their receptors?

Author Response

We want to thank the reviewer for the critical evaluation of our manuscript. We assessed data distribution by the normality tests provided by GraphPad Prism (D'Agostino & Pearson omnibus and Shapiro-Wilk normality test). We understand the reviewer’s concern about data distribution and therefore routinely check our statistical analyses with the nonparametric Mann-Whitney test comparing ranks, which were closely similar. In figures 1, 4, 5, 6, 8 and 9 we changed the statistical analysis to the suggested two-way ANOVA and used Dunnett’s post hoc test for multiple comparison where appropriate. When comparing two groups we used the unpaired T-test with Welch’s correction. We did not use sex as a variable in SF-Med since male and female cells did not show any sex difference under this condition. Only for SS-Med sex was used as an independent variable. In figure 4 the T-test and two-way ANOVA were both used, because here two different experiments are combined into one graph. First, we analyzed the effect of 10 nM and 100 nM dexa on mRNA expression (two-way ANOVA) and due to the lack of differences; we analyzed the effect of 1 nM dexa vs. control in a separate experiment (T-test). Furthermore, in figure 1C changing the statistical test eliminated the significance for testosterone, a result that we found difficult to interpret before, since the cells do not express androgen receptors. We want to thank the reviewer for these suggestions, which made our message clearer.

As mentioned above the FDLE cells do not express AR, which we showed previously. The AR inhibitor flutamide also did not affect the sex difference in Na+ transport (Schmidt et al., 2014). The cells do express the ER-beta, and in SS-Med the expression is sex-specific and female FDLE cells are more responsive to the stimulating effect of female sex steroids, as we have shown before (Kaltofen et al, 2015). But this did not affect the responsiveness to GCs, as shown in this manuscript. We included a short paragraph on this in the discussion of the manuscript and thank the reviewer for the suggestion.

Reviewer 3 Report

The authors of the manuscript entitled “Glucocorticoids equally stimulate epithelial Na+2 transport in male and female fetal alveolar cells” provide the results of a study aimed to evaluate the differences in the response of male and female fetal distal lung epithelial cells to glucocorticoids with regard to Na+ transport. The authors report that female and male alveolar cells exhibit no sex difference in responsiveness to GCs with regard to Na+ transport, and GR activity does not contribute to the higher Na+ transport in females. This type of information is rather interesting and could shed new light on the understanding of the absence of sex-specific effect of antenatal GC therapy. Although the manuscript has a limitation regarding the absence of protein expression data, the authors provided a good explanation of this disadvantage.

The article is well written, addresses all the questions raised and could be recommended for publication in the present form.

Author Response

We want to thank the reviewer for the encouraging revision of our manuscript.

Round 2

Reviewer 1 Report

The authors have responded to my critique by referencing previous work of their own and of others. They have not performed additional proof to rule out the basic concerns mentioned in my review. While I appreciate the thorough and well-written literature discussion, I think that scientific evidence rather than reference to the literature would be appropriate. Sentences that start with "we believe" have crawled into the literature as in the authors' response and are used to justify lack of scientific evidence. However, from my point of view the documentation of, for example, purity of the cells has to be checked in every study independent of whether the model is generally considered as a suitable model in the literature. What works in others' hands might not work in one's owns hands.